# Application of Advanced Hybrid Models to Identify the Sustainable Financial Management Clients of Long-Term Care Insurance Policy

You-Shyang Chen [1,*], Chien-Ku Lin [2], Jerome Chih-Lung Chou [3], Su-Fen Chen [4,5,*] and Min-Hui Ting [6]

1   College of Management, National Chin-Yi University of Technology, Taichung City 411, Taiwan
2   Department of Business Management, Hsiuping University of Science and Technology, Taichung City 412, Taiwan
3   Department of Information Management, Hwa Hsia University of Technology, New Taipei City 235, Taiwan
4   Department of Management and Information, National Open University, New Taipei City 247, Taiwan
5   Department of Business, National Open University, New Taipei City 247, Taiwan
6   Graduate Institute of Management, Chang Gung University, Taoyuan 333, Taiwan
*   Correspondence: yschen@ncut.edu.tw (Y.-S.C.); 10201a@gapps.nou.edu.tw (S.-F.C.)

**Abstract:** The rapid growth of the aging population and the rate of disabled people with physical and mental disorders is increasing the demand for long-term care. The decline in family care could lead to social and economic collapse. In order to reduce the burden of long-term care, long-term care insurance has become one of the most competitive products in the life insurance industry. In the previous literature review, few scholars engaged in the research on this topic with data mining technology, which was motivated to trigger the formation of this study and hoped to increase the different aspects of academic research. The purpose of this study is to develop the long-term insurance business from the original list of insurance clients, to predict whether the sustainable financial management clients will buy the long-term care insurance policies, and to establish a feasible prediction model to assist life insurance companies. This study aims to establish the classified prediction models of Models I~X, to dismantle the data with the percentage split and 10-fold cross validation, plus the application of two kinds of technology as feature selection and data discretization, for the data mining of twenty-three kinds of algorithms in seven different categories (Bayes, Function, Lazy, Meta, Misc, Rule, and Decision Tree) through the data collected from the insurance company database, and to select 20 conditional attributes and 1 decisional attribute (whether to buy the long-term insurance policy or not). The decision attribute is binary classification method for empirical data analysis. The empirical results show that: (1) the marital status, total number of policies purchased, and total amount of policies (including long-term care insurance) are found to be the three important factors affecting the decision attribute; (2) the most stable models are the advanced hybrid Models V and X; and (3) the best classifier is Decision Tree J48 algorithm for the study data used.

**Keywords:** long-term care; feature selection; data discretization; data mining; sustainable financial management

## 1. Introduction

In recent years, due to the progress of medical science and technology, the development of medicine, and the improvement of living standards, people's life expectancy is prolonged, and the population aging degree is serious. Fewer offspring increases the economic burden, and the caring problems of elderly life, disability, dementia, and chronic disease patients, resulting in relatively few elderly care resources. The rehabilitation course of injured patients with disability caused by accidents is quite long, and the cost is beyond the reach of ordinary double-income families. The huge medical expenses and long-term care costs have become social concerns and personal and family problems. All these problems show

the importance of long-term care needs in the future. Taking South Korea as an example, the proactive implementation of long-term care insurance in anticipation of population aging can provide the experience reference for countries with similar policy needs [1]. At present, the insurance companies in the industry are actively promoting the concept of sustainable long-term care insurance and related systems. However, due to the low awareness of crisis among the public, it shows that the promotion of long-term care insurance and system design needs to be strengthened. Due to physical and mental degeneration, elderly parents may gradually lose the ability for independent living and self-care, or they may suffer from chronic diseases; if so, they must be taken care of by family members or migrant workers, or even sent to nursing institutions. Based on an income evaluation of current double-income family, there is not much left over after deducting basic family life expenses, housing mortgages, car loans, child education expenses, and parent support payments; some cannot even make ends meet; thus, social problems arise. The cost of long-term care can also have a significant impact on families if they need the long-term care because of disability caused by accident or illness. Almost no one is immune to these future family problems. Thus, it is wise to plan for long-term care costs in advance, pay small amounts at ordinary times, be aware of the risk of using the insurance company in the event of accidents, and also save money if we are healthy in our old age. The long-term care premiums, similar to cancer insurance, are likely to rise over time. Thus, the issue of long-term care is a sustainable concern in financial supports. In particular, sustainable financial management can integrate environmental responsibility, reduce inequality, and increase energy efficiency and profits. Thus, it is preferable to drive sustainable policy development through the integration of environment and society for long-term success. With the rapid growth in the elderly population, the demand for long-term care has also continued to increase and the sustainability of funding care has been greatly challenged [2]. Long-term care needs arising from a rapidly ageing population can be supported through sustainable financial management. The government has the responsibility to build an effective long-term care system and to provide affordable long-term care services for families with disabilities. Long-term care insurance in the insurance industry can also reduce the financial stress of long-term care, disability, or illness. Thus, the issue of long-term care insurance is closed related to sustainable financial management. Through the promotion of insurance practitioner, the sustainable financial management client is able to be secured and reduce both the social and familial pressure from the long-term care problem. Finally, it can help enterprises increase the value, mission, and career life of insurance practitioners, and create a win–win–win situation for clients, companies, and individuals. Examples of past studies on data mining and long-term care insurance include the following review: research on government pension initiatives and their possible impact on an aging population [3], association between family loss and cognitive impairment [4], innovative insurance services in cloud computing [5], discussing customer churn for life insurance policies [6], and more. Although substantial studies have been performed on long-term care insurance, those further exploring long-term care insurance businesses through the behavior of customers purchasing insurance products are still critically lacking.

According to the conditional attributes of insurance clients, this study applies data mining tools to consolidate the data into reports, through machine learning, and identify the key factors making it possible to buy long-term care insurance. Through the mature and wide application of data mining technology in various fields, such as the prediction of immunization vaccine demand [7], big data analysis [8], prediction of bank loan risk [9], and the decision support system [10], especially the Naive Bayes classifier, K-Nearest Neighbor method (KNN), and Decision Tree (J48), etc., as well as other methods with good performance in other studies, the above problems can be solved. In this study, the conditional attributes of data status quo of the original clients' policy purchasing of life insurance companies in the industry, as well as data mining technology, have been used to discover important factors that might lead to customers repurchasing long-term care insurance. The data and rules have been summarized by modeling the prediction

accuracy and producing the decision tree graph. The research objectives can be summarized as follows: (1) to establish a classification prediction model, and find the most suitable classification model through data mining technology; (2) to find the best classifier through empirical study; (3) to find the rules and important key factors influencing the willingness to buy the long-term care insurance through the Decision Tree graph.

The remaining structure of the paper is as follows. Section 2 is a literature review of related studies, including the techniques of long-term care insurance and data mining. Section 3 describes the architecture of classification model. Section 4 is the empirical and analytical discussion, and Sections 5 and 6 are the management implications and research limitations of the empirical results, as well as the conclusions, discussions, and future research.

## 2. Literature Review

This section introduces the long-term care insurance, data mining, feature selection, data discretization, and classification algorithm.

### 2.1. Long-Term Care Insurance

Long-term care is for those who need the long-term care, and its services can include the treatment, prevention, diagnosis, rehabilitation, maintenance, support, and social services. Not only the individual, but the needs of caregivers should also be taken into account. Long-term care is the provision of long-term medical care, personal life care, and social services for those who are born or acquired without the ability to perform their daily life functions. Its service fields include social care and medical care. Long-term care is the provision of preventive, diagnostic, therapeutic, rehabilitative, supportive, and maintenance services to the entire population, including persons with disabilities and chronic diseases, at home, in institutions, and out of institutions. Long-term care insurance is aimed at people who are physically disabled or too old to take care of themselves in the hope of restoring, maintaining, or improving their daily life functions. The coverage scope of long-term care insurance is "when the insurant loses the self-care life ability, the long-term care insurance can pay regularly for the expense of long-term care", to provide the subsidy on the insurant nursing cost. Examples of related long-term care insurance research topics are: Japanese floods [11], sarcopenia [12], and population ageing [13].

### 2.2. Data Mining

Data mining is a tool or method for extracting knowledge from vast amounts of data. The process is similar to that of digging coal or mining, only the components are different. It is used to explore and analyze large amounts of data in automatic or semi-automatic ways, and use statistical, machine or deep learning methods to find meaningful relationships and rules. It is a continuous process of decision-making analysis, which seeks out valuable knowledge hidden in messy data from tedious data and provides it to all walks of life as a reference for decision-making. Using the modern computer technology as a tool, the huge data, through mining, cleaning, sorting to analyzing, it is to find the conditions, data, different rules, and useful information needed for research. The methods of data mining include the unsupervised, semi-supervised, supervised, and enhanced learning. Supervised learning includes classification, estimation, and prediction. Unsupervised learning includes clustering and association rule analysis. Many scholars compete in this research field; thus, the data mining technology has reached a stage of proficiency, and has been widely used in different fields, such as cloud systems [14], financial fraud detection [15], glasses sales [16], and many other fields.

### 2.3. Feature Selection

In the process of feature selection, the analysts, modeling tools, or algorithms will actively select or discard the attributes based on their usefulness to the analysis. Analysts may perform the feature engineering to add the features and remove or modify the existing

data, whereas machine learning algorithms typically score the data rows and verify their usefulness in the model. In short, the feature selection helps solving two problems: too much low-value data or too little high-value data. The key assumption for using the feature selection technique is that the training data contain many redundant or irrelevant features; therefore, removing these features will not result in information loss.

Feature selection is mostly applied to machine learning in combination with algorithms. According to specific evaluation indicators, it selects the highest value of discrimination power and effective identification rate from the original condition set. Ineffective indicators and those without critical impact are filtered. It simplifies the calculation, reduces the feature space and complexity, and helps to improve the evaluation efficiency. Feature selection is helpful to solve the problem of data asymmetry, and find the classification feature with the highest correlation ratio, and improve the feasibility of the model. Feature selection is commonly used in the classification and clustering of data-retrieval studies. Multiple sample attributes are selected for dimensionality reduction, and the features with high correlation and strong capability are selected as the research judgment data. Feature selection technology has been widely used in various research fields, such as machine learning [17], deep learning [18], and statistics [19].

### 2.4. Data Discretization

Data discretization refers to the segmentation of continuous data into discrete intervals. The discretization process is also described as the process of binning. There are two discretization methods. The first is to change the attribute to the classification level according to the expert's personal judgment, and the expert discretization is to facilitate the understanding of the result. The second is to ensure the accuracy of the value area, and the automatic discretization is to automatically cut the data by different equations.

Pal and Kar [20] pointed out that the data discretization is a pre-processing technology to mine the basic data from database, and the data generation rules after mining are very important. In machine learning, the data discretization and feature selection are important techniques for data pre-processing to improve the performance of algorithms in the high-dimensional data. The former focuses on converting the continuous attributes into discrete features, whereas the latter focuses on filtering out the unrepresentative features. Discretization is a continuous quantization process, and the continuous values are the most common. Rules that are short, compact, and precise bring the results that are easier to examine, use, compare, and reuse. The purpose of discretization is to find a concise form of data representation. As a kind of learning task, the data mining algorithm depends on the range and type of data.

### 2.5. Classification Algorithm

The algorithm for predicting the original unknown data after classification is the machine learning algorithm based on the rules obtained by analyzing the research data. It covers a large number of related statistical theories, the machine learning and inferential statistics are highly related, known as the statistical learning theory. As for algorithm design, the machine learning theory focuses on achievable and effective learning algorithms. Classification algorithm is to classify other data. From the abacus operation in the past, the computer operation is now used. Scientists write the designed formulas of mathematical problems into computer programs for linkage, and find the correct answer through operation. These formulas are called algorithms. Most researchers have widely used the algorithms in classification prediction, and found that the evaluation performance of decision tree is better than other classification tools.

Three commonly used and famous algorithms, namely, Decision Tree (trees-J48), Naive Bayes classifier and KNN (lazy-IBK), are selected as the representatives for illustration. The seven categories are briefly introduced as follows:

(1) Bayes: Bayes classifier is a practical application of Bayes' theorem, in which all features are assumed to be independent. According to Bayes' theorem, it calculates which

target has the highest probability of occurrence in the given data, and is used to make classification. Under the condition of large amount of data, Bayes classifier is a very useful model, which is simple and effective, and is not easy to produce the over-fitting. Naive Bayes classifier is a classifier based on the Bayes' theorem, which uses Bayes inference to calculate the probability of belonging to Classes, only suitable for classification. It is an effective tool for making predictions in uncertain situations, a statistical classification that uses the graphical models to represent the relationships among attributes, and can be used to calculate the possible values in the hope of achieving a complete and reasonable prediction. It mainly uses several possible prior and actual empirical assignments of the mother numbers to derive the post assignment of the mother numbers, and then calculates the possible values, hoping to achieve a perfect and reasonable prediction. It has the following characteristics: (a) Probability-type classification. (b) Calculate based on Bayes' theorem. (c) Assume that events are independent between features. (d) Application of automatic file classification. Bayes in this study include Bayes Net [21] and Naive Bayes classifier.

(2) Functions: Functions are a special category that contains a variety of classifiers that allow you to write the mathematical formulas in a natural and reasonable manner. The classifiers used in this study are (a) Logistic: polynomial logistic regression model classifier; (b) SGD: random gradient descent method for realizing various linear models; (c) SGD Text: random gradient descent method for realizing linear regression of text data; (d) Simple Logistic: the classifier establishing the linear logistic regression model; and (e) SMO: the implementation of minimum and best algorithm training supporting the vector machine John Platt sequence. Functions of this study include SMO [22], SGD [23], Simple Logistic [24], and SGD Text.

(3) Lazy: The lazy learning algorithm classifier does not need to use the training set for training, so the training time complexity is 0. K-Nearest Neighbor (KNN) algorithm is the most common application of lazy learning. The KNN method is an intuitive, nonparametric, concise, and effective classification method. It is regarded as the most basic test classifier and is easy to compare with complex classifiers. KNN classification identifies the training sample by assigning its category according to several adjacent sample points. In other words, the classification of training samples is determined by voting on several sample points adjacent to it, and the category of new sample point is determined by the largest number of votes. The similarity of test data and training data is compared with the nearest K data, and the classification is determined by vote. The K training values nearest to the test data will be outputted, and the test data with the largest number of classifications will determine the class of the test data. Among data mining technology classification methods, the simplest method is KNN. It is the non-parametric statistical method of the nearest K neighbors, for regression and classification [25]. This method is lazy learning. Instance-Based Learning is a kind of mechanical learning and non-parametric estimation method. Birds of a feather flock together. Therefore, as long as the nearest points to the unknown data are found from the training data, the data class can be judged to be the same as the class of nearest points. Lazy learning in this study includes IBK [26], K-Star [27], and LWL.

(4) Meta: As one of the integrated learning methods, Meta is most commonly used to solve the problem of a small amount of data. Its training data is in the form of set, which combines multiple classifiers. The training data of classifier can be randomly selected, or the weight of misclassified training data can be adjusted continuously, and then the new classifier can be generated with it. Experimental test analysis is performed using the following four common algorithms: Ada Boost M1, Bagging, Stacking, and Vote. Meta of this study includes Stacking, Vote, and AdaBoost M1 [28].

(5) Misc: Misc is an application of packaging classification technology. In this study, the Input Mapped Classifier is used for experimental research. Its characteristic is to solve the incompatible problem of test and training data through the correspondence between the training data of the classifier and the input test instance structure. At-

tributes that the classifier has not seen before are received as the missing value. You can train a new classifier or load an existing classifier from a file.

(6) Rules: Rule is a relatively simple classification technique that uses a set of judgment rules to classify the data. Coverage and accuracy are used to measure the quality of classification rules. Rule classifier has two important features, which are mutually exclusive rules and exhaustive rules. In order to establish a rule-based classifier, the process needs to establish a set of rules. There are two methods to analyze the classification rules: (a) direct method and (b) indirect method. The following five common algorithms are applied in this study: Decision Table [29], JRip, OneR, PRAT, and ZeroR.

(7) Decision Tree: Decision Tree will generate a tree according to the training data and predict the new samples according to the trained rules. Decision tree algorithm can use different ways to evaluate the quality of branches (chaos), such as Information gain [30], Gain ratio [31], and Gini index [32]. According to the training data, it shall find the appropriate rules, and finally generate a rule tree to make all decisions, and its purpose is to make each decision to maximize the information gain. Decision tree is generally generated from top to down. Each event and natural state may develop two or more different events, and form different results, to branch the decision into a graph, similar to a tree; thus, it is called the decision tree. There are many ways to divide it, and the end goal is the same: from the bottom of the root up to the leaf, which is the path and rule. However, it can have two or more branches. Application of decision trees uses the decision trees or decision models, to create plans to reach goals by support tools. It can explicitly express the decision-making process, which is often used in decision analysis. The special structure of tree is an algorithm display method. Trees in this study mainly include J48 [33,34], LMT [35,36], and REP Tree [37,38] classifiers.

## 3. Research Method

This section mainly introduces the framework of classification models proposed in this study and its research steps, as well as the actual case analysis data, to identify the potential clients of long-term care insurance policies by data mining predictive model technology.

### 3.1. Research Framework

This study takes the data of a leading life insurance company in Taiwan as an empirical study. Firstly, it confirms the research project, plans the condition data required for the research, and obtains the condition data of life insurance clients from August 1997 to March 2021 from the database of anonymous abbreviation A-Life Insurance Company based on principle of personal information. Additionally, the data are cleaned up and the data table is prepared. Then, the code is set up and the spreadsheet is prepared. Finally, the file format is converted into the format used in mining and prediction in CSV file. The decision tree model is constructed by binary classification method, and the rules are found by algorithm and explained. It is applied in the decision making of association analysis of conditional data to help companies and insurance salesmen screening and finding out the valid client list through the decision analysis results, so that the salesmen can achieve twice the result with half the effort in promoting the insurance business of clients buying the long-term care insurance policies. The classification model algorithm in this study mainly consists of 10 steps, and the research framework is shown in Figure 1.

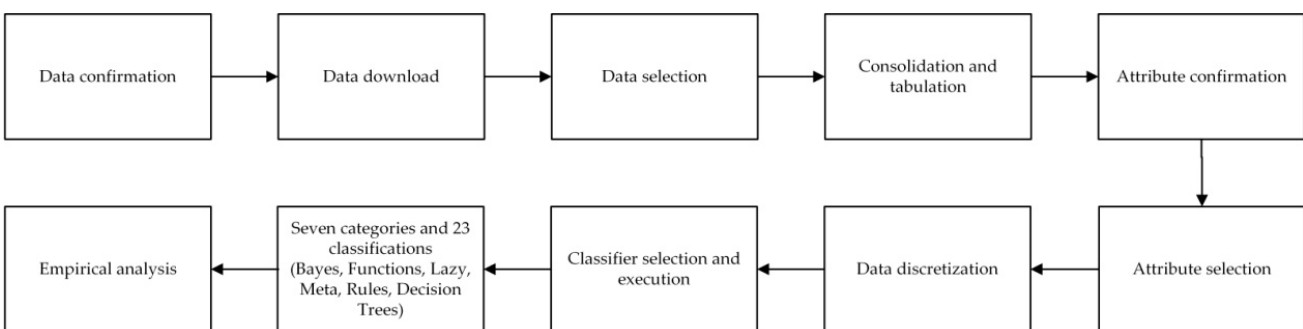

**Figure 1.** Research framework.

*3.2. Research Steps and Examples of Classification Model Algorithm*

This study proposes an effective long-term care insurance client classification prediction model, which uses a real case and is divided into 10 key steps to be described in detail, as below.

Step 1. Data confirmation: In the wake of the Silver Tsunami, the world is concerned about the consequences of long-term care problem. Combined with the data of life insurance companies in the industry, it is proposed to purchase the long-term care insurance policy to solve the economic impact brought by the unknown huge expenditure, and to purchase the preventive long-term care insurance to achieve the risk transfer through small expenditure. Data mining is carried out in the purchasing data of life insurance companies in the past to find out the conditional attribute of clients that are likely to purchase the long-term care insurance products, so as to provide a reference for decision makers when making decisions. This is the primary task of knowledge mining.

Step 2. Data download: In this study, the data are collected through various conditional attribute data from the life insurance company database, and then gradually sorted and filtered, which becomes the main format and complete research data. The information is downloaded from the insurance contents of the named A-Life Insurance Company from August 26 1997 to March 31 2021 for an extensive data examination over a long period of time. A total of 473 pieces of data are sorted, filtered, and integrated into the required conditional data of this study.

Step 3. Data selection: Filter the contents of data, refer to the expert's opinions, select the data for sorting and summary, and consolidate them into Excel.

Step 4. Consolidation and tabulation: The downloaded insurance data will be sorted out in Excel. Firstly, the attribute data will be imported into the table, the incomplete data will be deleted through filtering, and the details will be checked and sorted out. Finally, the format will be converted into CSV files required by the prediction tool.

This step will sort the sample data and convert the format as follows:

(1) Data filtering and sorting: delete the incomplete, redundant and similar conditional data, maintain the data integrity and effectiveness, and improve the efficiency of machine learning.

(2) Data integration: follow the above Step 1, correctly and carefully check the data, and confirm whether the set columns (attribute conditions in Excel) are correct, such as: correctness and completeness of the data as gender, marital status and educational background. In the data mining, there is a positive relationship between data quality and quantity, which plays a decisive role in the research and exploration, analysis and result.

(3) Conversion of research data format: convert the format of the integrated research data file into the format (CSV file) that meets the requirements of subsequent data mining and prediction, so as to facilitate the data mining and analysis tools to explore the data for prediction and analysis.

Step 5. Attribute confirmation: By filtering out similar or redundant data from insurance client data, 21 research attributes, including 20 conditional attributes and one decision attribute, are selected based on the expert's opinions.

(1)  In total, 20 conditional attributes: gender, educational background, nature of work, marital status, whether the spouse is on-the-job, family salary structure, total number of purchased insurance policies, effective number of insurance policies, time interval to the first purchase, life insurance amount, major disease insurance amount, payment type (yearly, half-year, seasonally or monthly), total premium of annual effective policies of oneself, whether to continue to pay the insurance premium, payer, total amount of life insurance (including major disease insurance)/TWD10,000, total insurance amount (including long-term care and disability insurance), whether there is an investment policy, whether to buy a new policy, whether there is a transfer, and whether to buy the long-term care insurance policy products, are selected according to the expert's opinions and the machine.

(2)  One decision attribute: this study takes whether to purchase the long-term care insurance policy as the decision attribute, and the decision attribute in this study is expressed in the text form. The category data type adopts the most commonly used binary classification method, "Y" for Yes and "N" for No.

(3)  Coding: since the research tool is in English, the attributes in this study are coded E1, E2, E3, ..., E21, as detailed in Table 1; it also explains the meaning and text of the attribute, among which the important decision attribute is whether to buy the long-term care insurance policy (E21).

Step 6. Attribute selection: One decision attribute: Whether to buy the long-term care insurance policy (E21), and 473 research data by binary classification method are used for machine attribute selection. The data type of decision attribute in this study is the category data.

Step 7. Data discretization: In this step, the experimental data are pre-processed discretely by machine data.

Step 8. Classifier selection and execution: In this step, two different data disassembly methods are used to evaluate the performance of the algorithm modeling: percentage split and cross validation, as described below. (1) Percentage split: The datasets in the experiment will divide all data into the training sets and test sets, accounting for about 67% and 33% of the normal proportion. The random classification in machine learning is used to train the training sub-set, and then the remaining testing sub-set is used to verify the model. According to the test results, the prediction models are evaluated to find out the most appropriate model. In this study, the data are divided into six ratios: 50/50, 60/40, 70/30, 80/20, 90/10, and 67/33 for training/testing to verify the effectiveness of the model. (2) Cross validation: The validation method adopts 10-fold cross validation. The data set is divided into 10 folds, 9 of which are alternately used as the training data and 1 as the test data, and the model is verified by analyzing the results to find out the optimal model.

Step 9. Empirical analysis: The algorithm is selected from the data mining tools to conduct the data mining, and the model with the best performance is selected to produce the tree graph by the decision tree. In the process of data visualization, the tree graph is summarized for research interpretation and application, making it easy to understand and read the data analysis results. The single and hybrid prediction classification models established in this study are shown in Table 2.

**Table 1.** Description of attribute data sheet.

| SN | Code | Data Type | Attribute | Formal Specification |
|----|------|-----------|-----------|----------------------|
| 1 | E1 | Category | Gender (m for male, f for female) | Text |
| 2 | E2 | Category | Educational background (G: below junior high school, H: high school, C: junior college, U: university, M: master, P: PhD) | Text |
| 3 | E3 | Category | Nature of work (Y for office work, N for non-office work) | Text |
| 4 | E4 | Category | Marital status (Y for married, N for unmarried) | Text |
| 5 | E5 | Category | Whether the spouse is on-the-job (Y for on-the-job, N for not on-the-job) | Text |
| 6 | E6 | Category | Family salary structure (S for single income, D for double income) | Text |
| 7 | E7 | Scope data | Total number of purchased insurance policies (indicate by number) | Number |
| 8 | E8 | Scope data | Effective number of insurance policies (indicate by number) | Number |
| 9 | E9 | Scope data | Time interval to the first purchase (0~35 years, indicate by number) | Number |
| 10 | E10 | Scope data | Life insurance amount (indicate by number) | Number |
| 11 | E11 | Scope data | Major disease insurance amount (indicate by number) | Number |
| 12 | E12 | Scope data | Payment type (y for yearly, h for half-year, s for seasonally or m for monthly) | Text |
| 13 | E13 | Scope data | Total premium of annual effective policies of oneself (indicate by number) | Number |
| 14 | E14 | Scope data | Whether to continue to pay the insurance premium (Y for yes, N for no) | Text |
| 15 | E15 | Scope data | Payer (I for oneself, S for spouse, P for parent, C for child) | Text |
| 16 | E16 | Scope data | Total amount of life insurance (including major disease insurance)/TWD 10,000 (TWD 10,000 as the unit) | Number |
| 17 | E17 | Scope data | Total insurance amount (including long-term care and disability insurance indicate by number) | Number |
| 18 | E18 | Category | Whether there is an investment policy (Y for yes, N for no) | Text |
| 19 | E19 | Category | Whether to buy a new policy (Y for yes, N for no) | Text |
| 20 | E20 | Category | Whether there is a transfer (Y for yes, N for no) | Text |
| 21 | E21 | Category | Whether to buy the long-term care insurance policy products (Y for yes, N for no) | Text |

**Table 2.** Single and hybrid prediction classification model sheet Evaluation method.

| Model | I | II | III | IV | V | VI | VII | VIII | IX | X |
|-------|---|----|-----|----|----|----|-----|------|----|----|
| Percentage split | V | V | V | V | V | | | | | |
| Cross validation | | | | | | V | V | V | V | V |
| Algorithms (23) | V | V | V | V | V | V | V | V | V | V |
| Data discretization | | | V | (1) V | (2) V | | | V | (1) V | (2) V |
| Attribute selection | | V | | (2) V | (1) V | | V | | (2) V | (1) V |

Note: 1. "V" indicates the adopted method. 2. Models IV, V, IX, and X use two techniques of attribute selection and data discretization, wherein (1) and (2) represent the sequence. An example of Model V: attribute selection first and then data dispersion.

Step 10. Result analysis: According to a variety of experimental analysis, different kinds of mining techniques are used to evaluate each classification algorithm, evaluate the accuracy in the classification prediction model, select the best prediction model, and find out the conditional attributes that affect them, and through the experiment, to establish a prediction model for effect prediction. Based on the experimental results, this paper explains the research results and puts forward some suggestions for commercial insurance decision-making.

## 4. Theoretical and Empirical Analysis

According to the empirical results of the hybrid classification model established in the previous section, the accuracy is compared and analyzed. The research data are processed by data technology, and the accuracy of classification model established in this study is evaluated and analyzed with empirical research purpose and research method. The first subsection is the experimental analysis, and the second subsection is the experimental results and findings. The classification algorithm is used to evaluate the accuracy performance, and the results are analyzed. According to the experimental output, the important indicators are established by the rule of decision tree. For example: in which conditions the client would buy the long-term care insurance, and find out the important factors, to improve the turnover ratio, find solutions to the problem of long-term care and disability, and alleviate future financial fears of family caregivers and the society.

### 4.1. Experimental Analysis

The classification algorithm is analyzed in this section. The prediction model uses the life insurance company's client data, examining whether to buy the long-term care insurance policies as the decision attribute, and adopts seven categories of 23 classifiers for the binary decision attribute by percentage split (5 models) and 10-fold cross validation (5 models); in total, 10 models are used for implementation, analysis, evaluation and calculation of performance. The two stages are explained in order, as follows.

Stage 1. Model implementation: Three core directions are addressed for this implementing function.

(1)     Percentage split attribute performance evaluation: Models I~V.

Model I: Without data discretization, without attribute selection
Model II: Without data discretization, with attribute selection
Model III: With data discretization, without attribute selection
Model IV: Data discretization before attribute selection
Model V: Attribute selection before data discretization

(2)     Cross validation attribute performance evaluation: Models VI~X.

Model VI: Without data discretization, without attribute selection
Model VII: Without data discretization, with attribute selection
Model VIII: With data discretization, without attribute selection
Model IX: Data discretization before attribute selection
Model X: Attribute selection before data discretization

(3)     Analysis of decision trees graph for decision-making purposes.

Stage 2. Model analysis and performance evaluation: Three key directions are also addressed for this implementing function.

(1)     Percentage split attribute performance evaluation: for whether to purchase the long-term care insurance, five models (Models I, II, III, IV, and V) are established among 23 classifiers of seven categories by using percentage split method, to select the classifiers with better accuracy. Taking the best Model V as an example, the description is shown in Table 3.

**Table 3.** Accuracy statistical table of Model V: attribute selection before data discretization by percentage split.

| Category | Classifier | 50/50 | 60/40 | 70/30 | 80/20 | 90/10 | 67/33 |
|---|---|---|---|---|---|---|---|
| Bayes | Bayes Net | 80.5085 | 79.3651 | 85.2113 | 82.1053 | 80.8511 | 77.5641 |
| | Naïve Bayes | 80.5085 | 79.3651 | 85.2113 | 82.1053 | 80.8511 | 77.5641 |
| Functions | Logistic | 85.1695 | 84.1270 | 85.2113 | 82.1053 | 80.8511 | 83.9744 |
| | SGD | 80.5085 | 83.5979 | 84.5070 | 82.1053 | 82.9787 | 83.3333 |
| | SGD Text | 69.9153 | 68.2540 | 66.9014 | 64.2105 | 72.3404 | 66.0256 |
| | Simple Logistic | 85.1695 | 84.1270 | 85.2113 | 82.1053 | 80.8511 | 83.9744 |
| | SMO | 80.5085 | 79.3651 | 84.5070 | 82.1053 | 82.9787 | 77.5641 |
| Lazy | IBk | 85.1695 | 84.1270 | 85.2113 | 82.1053 | 80.8511 | 83.9744 |
| | KStar | 85.1695 | 82.0106 | 83.0986 | 78.9474 | 78.7234 | 82.0513 |
| | LWL | 83.0508 | 80.4233 | 82.3944 | 76.8421 | 78.7234 | 80.1282 |
| Meta | AdaBoostM1 | 78.3898 | 76.7196 | 75.3521 | 70.5263 | 76.5957 | 75.0000 |
| | Bagging | 85.1695 | 84.1270 | 85.2113 | 82.1053 | 80.8511 | 77.5641 |
| | Stacking | 69.9153 | 68.2540 | 66.9014 | 64.2105 | 72.3404 | 66.0256 |
| | Vote | 69.9153 | 68.2540 | 66.9014 | 64.2105 | 72.3404 | 66.0256 |
| Misc | Input Mapped Classifier | 69.9153 | 68.2540 | 66.9014 | 64.2105 | 72.3404 | 66.0256 |
| Rules | Decision Table | 83.8983 | 82.5397 | 85.2113 | 82.1053 | 80.8511 | 82.0513 |
| | JRip | 84.7458 | 84.1270 | 84.5070 | 81.0526 | 80.8511 | 83.9744 |
| | OneR | 80.5085 | 79.3651 | 78.1690 | 74.7368 | 78.7234 | 77.5641 |
| | PART | 85.1695 | 84.1270 | 85.2113 | 82.1053 | 80.8511 | 83.9744 |
| | ZeroR | 69.9153 | 68.2540 | 66.9014 | 64.2105 | 72.3404 | 66.0256 |
| Trees | J48 | 85.1695 | 84.1270 | 85.2113 | 82.1053 | 80.8511 | 83.9744 |
| | LMT | 85.1695 | 84.1270 | 85.2113 | 82.1053 | 80.8511 | 83.9744 |
| | REP Tree | 85.1695 | 84.1270 | 85.2113 | 82.1053 | 80.8511 | 83.9744 |

Unit: (%).

The accuracy evaluation of Model V: attribute selection before data discretization is explained as follows.

(a) Bayes: Bayes Net-lowest 77.5641% and highest 85.2113%; Naive Bayes-lowest 77.5641% and highest 85.2113%. Bayes Net and Naive Bayes' lowest 77.5641% and highest 85.2113% are the same.

(b) Functions: Logistic-lowest 80.8511% and highest 85.2113%; SGD-lowest 80.5085% and highest 84.5070%; SGD Text-lowest 64.2105% and highest 72.3404%; Simple Logistic-lowest 80.8511% and highest 85.2113%; SMO-lowest 77.5641% and highest 84.5070%. In this category, SGD Text has the lowest 64.2105%, and Logistic and Simple Logistic have the highest 85.2113%.

(c) Lazy: IBk-lowest 80.8511% and highest 85.2113%; K Star-lowest 78.7234% and highest 85.1695%; LWL-lowest 76.8421% and highest 83.0508%. In this category, LWL has the lowest 76.8421% and IBk has the highest 85.2113%.

(d) Meta: Ada Boost M1-lowest 70.5263% and highest 78.3898%; Bagging-lowest 77.5641% and highest 85.2113%; Stacking and Vote classifiers have the same lowest 64.2105% and highest 72.3404%, which are unsatisfactory. In this category, Stacking and Vote have the lowest 64.2105% and Bagging has the highest 85.2113%.

(e) Misc: Input Mapped Classifier has the lowest 64.2105% and highest 72.3404%, not good enough.

(f) Rules: Decision Table-lowest 80.8511% and highest 85.2113%; JRip-lowest 80.8511% and highest 84.7458%; OneR-lowest 74.7368% and highest 80.5085%; PART-lowest 80.8511% and highest 85.2113%; ZeroR-lowest 64.2105% and highest 72.3404%, poor performance. In this category, ZeroR has the lowest 64.2105%, and the evaluation values of Decision Table and PART are equally good at 85.2113%.

(g) Trees: J48, LMT, and REP Tree have the lowest 80.8511% and highest 85.2113%, and other proportion measurements (marked in *italics*). The measurement values of each proportion of these three classifiers are the same, and all values are in the range of 80.8511~85.2113%. Among the seven categories of Model V, the trees show the most stable performance.

In summary, the prediction values of Bayes Net, Naive Bayes, Logistic, Simple Logistic, IBk, Bagging, Decision Table, PART, J48, LMT, and REP Tree are 85.2113%, which are the good classifiers with same highest value in the study data used. Importantly, the lowest accuracy, highest accuracy and difference values of the seven categories of Model V by percentage split are shown in Table 4; at the same time, Figure 2 shows the bar chart of lowest, highest, and difference values of seven categories of Model V.

**Table 4.** Comparison table of highest and lowest values of Model V by percentage split.

| Category | Bayes | Functions | Lazy | Meta | Misc | Rules | Trees |
|---|---|---|---|---|---|---|---|
| Highest value | 85.2113 | 85.2113 | 85.2113 | 85.2113 | 72.3404 | 85.2113 | 85.2113 |
| Lowest value | 77.5641 | 64.2105 | 76.8421 | 64.2105 | 64.2105 | 64.2105 | 80.8511 |
| Difference value | 7.6472 | 21.0008 | 8.3692 | 21.0008 | 8.1299 | 21.0008 | 4.3602 |

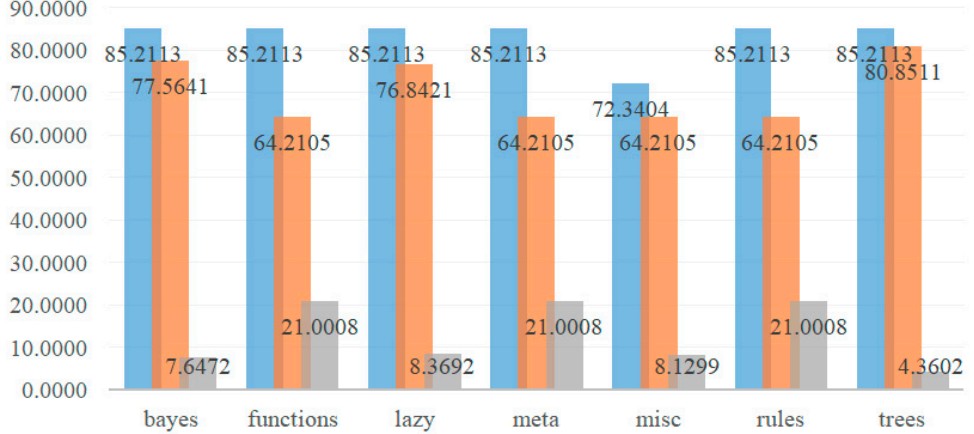

**Figure 2.** Bar chart of lowest, highest and difference values of seven categories of Model V.

It can be seen from Tables 3 and 4 and Figure 2, the lowest and highest accuracy performance and difference value of prediction Model V and algorithm are compared and analyzed as follows.

(a) The proportion performance value shows that from the evaluation value presented by the three classifiers of trees (J48, LMT, and REP Tree), the accuracy is relatively stable on the whole, with 85.2113% as the highest, 80.8511% as the lowest, and the difference value 4.3602% is the smallest.

(b) In the seven categories, the accuracy of SGD Text of Functions, Stacking and Vote of Meta, Input Mapped Classifier of Misc, and ZeroR classifier of Rules is not good, which is the lowest evaluation value (64.2105%) in the percentage split.

(c) Among the seven categories in summary table, the highest 85.2113% is distributed in Bayes Net and Naive Bayes of Bayes, Logistic and Simple Logistic of Functions, IBk of Lazy, Bagging of Meta, Decision Table and PART classifier of Rules and J48, and LMT and REP Tree of Trees, and all distributed in 70/30 (training value/test value).

(d) For the accuracy of Model V: attribute selection before data discretization by percentage split, Functions and Meta and Rules have the lowest value of 64.2105%, the highest value of 85.2113%, and the largest difference value of 21.0008%.

According to the comparative analysis results of Models I~V, the summary is described below.

(a)  Model I: The results of Logistic, PART, and J48 classifiers are the top three classifiers in the study.

(b)  Model II: The results of Bagging, LMT, REP Tree, J48, and PART are the top five.

(c)  Model III: Logistic, SMO, Bagging, JRip, Simple Logistic, J48, and LMT have higher values and better results.

(d)  Model IV: IBk, REP Tree, K Star, Bagging, Decision Table, JRip, PART, and J48 classifiers have higher values and better results.

(e)  Model V: Bayes Net, Naive Bayes, Logistic, Simple Logistic, IBk, Bagging, Decision Table, PART, J48, LMT, and REP Tree classifiers have higher values and better results.

In total, the analysis of highest value difference of Model I~Model V is showed in Table 5. From Table 5, there are 10 core resolutions identified, as follows:

(a)  Difference of Model II and Model I: except for 100% of Functions, others have little difference.

(b)  Difference of Model III and Model I: except for 100% of Functions, Bayes, Lazy, and Meta of Model III are better than Model I, Misc has same performance in these two models; Model III is better than Model I.

(c)  Difference of Model IV and Model I: Bayes and Lazy of Model IV are better than Model I; in addition to this, Model I is better.

(d)  Difference of Model V and Model I: only Bayes and Lazy of Model V are better than Model I, and the others of Model I is better.

(e)  Difference of Model III and Model II: Bayes, Functions, Rules, and Trees of Model III are better than Model II, whereas Lazy, Misc, and Meta are the same. Model III is better than Model II.

(f)  Difference of Model IV and Model II: Bayes, Functions, Lazy, and Trees of Model IV are better than Model II, whereas Meta, and Rules of Model IV are poorer; Model IV is better than Model II.

(g)  Difference of Model V and II: Model V is better than Model II in Bayes, Functions, and Lazy, two models are the same in Misc, and Model II is better than Model V in Meta, Rules, and Trees.

(h)  Difference of Model IV and Model III: except Lazy of Model III is poorer, and Misc is the same in two models, Bayes, Functions, Meta, Rules, and Trees are better in Model III than Model IV.

(i)  Difference of Model V and Model III: Bayes, Functions, Meta, Rules, and Trees are better in Model III than Model V, Lazy of Model III is poorer, and Misc is the same in two models.

(j)  Difference of Model V and Model IV: Bayes and Functions of Model V are better than Model IV, Misc, and Rules have the same performance in two models.

The analysis result of classifiers with higher accuracy in Models I~V is shown in Table 6. J48 is used five times in total, whereas Bagging and PART are used four times, respectively; Logistic, LMT, and REP Tree are used three times, respectively; Simple Logistic, IBk, Decision Table, and JRip are used two times, respectively. It is therefore clear that J48, Bagging, and PART are the top three classifiers with better performance times for the study data used.

**Table 5.** Comparison table of highest value difference of Model I~Model V.

| Category | Bayes | Functions | Lazy | Meta | Misc | Rules | Trees | Highest Value |
|---|---|---|---|---|---|---|---|---|
| Model I | 84.2105 | 100.0000 | 78.7234 | 88.4211 | 72.3404 | 93.6170 | 88.8889 | 100.0000 |
| Model II | 82.3944 | 76.5957 | 85.1064 | 89.3617 | 72.3404 | 85.7143 | 87.2340 | 89.3617 |
| Model III | 87.2340 | 89.3617 | 85.1064 | 89.3617 | 72.3404 | 88.7324 | 88.4615 | 89.3617 |
| Model IV | 85.1064 | 85.1064 | 88.7324 | 86.3158 | 72.3404 | 85.2113 | 88.0282 | 88.7324 |
| Model V | 85.2113 | 85.2113 | 85.2113 | 85.2113 | 72.3404 | 85.2113 | 85.2113 | 85.2113 |
| **Model comparison Difference value** | | | | | | | | |
| II and I | −1.8161 | −23.4043 | 6.3830 | 0.9406 | 0.0000 | −7.9027 | −1.6549 | −10.6383 |
| III and I | 3.0235 | −10.6383 | 6.3830 | 0.9406 | 0.0000 | −4.8846 | −0.4274 | −10.6383 |
| IV and I | 0.8959 | −14.8936 | 10.0090 | −2.1053 | 0.0000 | −8.4057 | −0.8607 | −11.2676 |
| V and I | 1.0008 | −14.7887 | 6.4879 | −3.2098 | 0.0000 | −8.4057 | −3.6776 | −14.7887 |
| III and II | 4.8396 | 12.7660 | 0.0000 | 0.0000 | 0.0000 | 3.0181 | 1.2275 | 0.0000 |
| IV and II | 2.7120 | 8.5107 | 3.6260 | −3.0459 | 0.0000 | −0.5030 | 0.7942 | −0.6293 |
| V and II | 2.8169 | 8.6156 | 0.1049 | −4.1504 | 0.0000 | −0.5030 | −2.0227 | −4.1504 |
| IV and III | −2.1276 | −4.2553 | 3.6260 | −3.0459 | 0.0000 | −3.5211 | −0.4333 | −0.6293 |
| V and III | −2.0227 | −4.1504 | 0.1049 | −4.1504 | 0.0000 | −3.5211 | −3.2502 | −4.1504 |
| V and IV | 0.1049 | 0.1049 | −3.5211 | −1.1045 | 0.0000 | 0.0000 | −2.8169 | −3.5211 |

**Table 6.** Statistical table of classifiers with higher accuracy in Models I~V.

| Category | Classifier | Model I | Model II | Model III | Model IV | Model V | Total Times | Rank |
|---|---|---|---|---|---|---|---|---|
| Bayes | Bayes Net | | | | | V | 1 | |
| | Naïve Bayes | | | | | V | 1 | |
| Functions | Logistic | V | | V | | V | 3 | |
| | Simple Logistic | | | V | | V | 2 | |
| | SMO | | | V | | | 1 | |
| Lazy | IBk | | | | V | V | 2 | |
| | KStar | | | | | V | 1 | |
| Meta | Bagging | | V | V | V | V | 4 | 2 |
| Rules | Decision Table | | | | V | V | 2 | |
| | JRip | | | V | | V | 2 | |
| | PART | V | V | | V | V | 4 | 2 |
| Trees | J48 | V | V | V | V | V | 5 | 1 |
| | LMT | | V | V | | V | 3 | |
| | REP Tree | | V | | V | V | 3 | |

From summary of Table 6, the following six meaningful tentative consequences are determined, and they can be referenced for the respective purposes of academics and practitioners in the future.

(a) According to the above data, Model III with data discretization is a better model. This may be due to the fact that all data are the information type, and the accuracy is high and good after discretization.

(b) The sequence technology of Model IV and Model V shows that the accuracy of Model V, with attribute selection before data discretization, is more average, but the accuracy of Model IV, with data discretization before attribute selection, is higher.

(c) Compared with Model III with data discretization, the prediction value of Model II with attribute selection is slightly poorer to that of Model III.

(d) In both Model III and Model IV with data discretization technology, the prediction value of Model IV is not much different from that of Model III, which may be because these attributes are all very important. However, Model IV also with attribute selection may have some attributes missing and the evaluated value slightly decreases.

(e) As explained in (4), for Model I and Model II, the prediction value of Model I without attribute selection and without data discretization is slightly higher than that of Model II with attribute selection.

(f) In Table 5, Model V is the most stable in terms of classification accuracy.

(2) Cross validation attribute performance evaluation: Among 23 classifiers of seven categories, 5 models (Models VI, VII, VIII, IX, and X) are established by using cross validation method, in order to find out the prediction classifier with best performance. Taking the best Model X as an example, the description is as below.

The accuracy analysis result of Model X: attribute selection before data discretization is showed in Table 7.

**Table 7.** Cross validation statistical table of Model X purchasing the disability care insurance-attribute selection before data discretization.

| Category | Classifier | Cross Validation |
|---|---|---|
| Bayes | Bayes Net | 87.3150 |
| | Naïve Bayes | 87.3150 |
| Functions | Logistic | 86.6808 |
| | SGD | 85.2008 |
| | SGD Text | 68.4989 |
| | Simple Logistic | 87.3150 |
| | SMO | 84.9894 |
| Lazy | IBk | 87.3150 |
| | KStar | 84.5666 |
| | LWL | 81.3953 |
| Meta | AdaBoostM1 | 79.7040 |
| | Bagging | 85.8351 |
| | Stacking | 68.4989 |
| | Vote | 68.4989 |
| Misc | Input Mapped Classifier | 68.4989 |
| Rules | Decision Table | 87.1036 |
| | JRip | 86.6808 |
| | OneR | 82.6638 |
| | PART | 87.3150 |
| | ZeroR | 68.4989 |
| Trees | J48 | 87.3150 |
| | LMT | 87.3150 |
| | REP Tree | 86.2579 |

Unit (%).

There are the following seven key points addressed for the used data from Table 7 to conclude the reported empirical results.

(a) Bayes: Bayes Net and Naive Bayes are the same 87.3150%.
(b) Functions: Simple Logistic of 87.3150% is in high performance, whereas Logistic of 86.6808%, SGD of 85.2008%, SMO of 84.9894%, and SGD Text of 68.4989% are in low performance.
(c) Lazy: IBk of 87.3150% is in high performance, whereas K Star of 84.5666% and LWL of 81.3953% are in low performance.
(d) Meta: Bagging of 85.8351% is in high performance, whereas Ada Boost M1 of 79.7040%, and Stacking and Vote of 68.4989% are in low performance.
(e) Misc: Input Mapped Classifier of 68.4989% is in low performance.
(f) Rules: PART of 87.3150% is in high performance, and Decision Table of 87.1036% is in the second high performance, whereas JRip of 86.6808%, OneR of 82.6638%, and ZeroR of 68.4989% are in low performance.

(g)  Trees: J48 and LMT of 87.3150% are in high performance, and REP Tree of 86.2579%, with good overall performance.

In summary, the accuracy of Bayes Net, Naive Bayes, Simple Logistic, IBk, PART, J48, and LMT is the same (87.3150%), which are the classifiers with good performance. The highest accuracy, lowest accuracy and difference values of the seven categories of Model X by cross validation are shown in Table 8.

**Table 8.** Comparison table of highest and lowest values of Model X by cross validation.

| Category | Bayes | Functions | Lazy | Meta | Misc | Rules | Trees |
|---|---|---|---|---|---|---|---|
| Highest value | 87.3150 | 87.3150 | 87.3150 | 85.8351 | 68.4989 | 87.3150 | 87.3150 |
| Lowest value | 87.3150 | 68.4989 | 81.3953 | 68.4989 | 68.4989 | 68.4989 | 86.2579 |
| Difference value | 0.0000 | 18.8161 | 5.9197 | 17.3362 | 0.0000 | 18.8161 | 1.0571 |

Unit (%).

For summary, from Tables 7 and 8, it shows the highest, lowest, and difference values of prediction Model X and the algorithm.

(a)  Cross validation performance values: the overall performance of evaluation values presented on the three classifiers of Trees (J48, LMT, and REP Tree) is relatively stable; J48 and LMT of 87.3150% are the highest value, and REP Tree of 86.2579% is the lowest value. The difference between the highest and lowest values is 1.0571%.

(b)  In Bayes, the highest and lowest accuracy are 87.3150%, and the difference of 0.0000% is the least, showing excellent performance.

(c)  In Lazy, IBk, K Star and LWL classifiers in order is 87.3150%, 84.5666% and 81.3953%, and the difference value is 5.9197%.

(d)  The accuracy of Functions and Rules on other classifiers is 87.3150~85.2008% and 87.3150~82.6638%, with good performance. In addition, both Functions-SGD Text and Rules-ZeroR show the lowest accuracy of 68.4989%, and the difference between the highest and lowest values of these two classifiers is 18.8161%, which shows the largest difference, and the overall performance is not good.

(e)  Among the seven categories, the highest accuracy rate is 87.3150%. It is generally presented in Bayes Net and Naive Bayes of Bayes, Simple Logistic of Functions, IBk of Lazy, PART of Rules and J48 and LMT of Trees.

(f)  Among the seven categories, the lowest accuracy of 68.4989% occurs simultaneously in SGD Text of Functions, Stacking and Vote of Meta, Input Mapped Classifier of Misc, and ZeroR of Rules.

According to the analysis results of Models VI~IX, the summary is described as below.

(a)  Model VI: The results of Logistic, J48 and PART classifiers are good and ranked top three.

(b)  Model VII: IBk, KStar, Bagging, JRip, J48, and LMT present higher values and are the good classifier.

(c)  Model VIII: SGD, LMT, SMO, PART, Bagging, and Simple Logistic classifiers are ranked top.

(d)  Model IX: J48, PART, IBk, Logistic, LMT, and Bagging classifiers are ranked top.

(e)  Model X: The accuracy of Bayes Net, Naive Bayes, Simple Logistic, IBk, PART, J48, and LMT is the same, which are the good classifiers.

Consequently, the analysis of highest value difference of Model VI~Model X shows in Table 9. From Table 9, 10 important results are reported as follows:

(a)  Difference of Models VII and VI: except the comparison value difference of functions and Lazy exceeds 22%, other performances are improved slightly.

(b)  Difference of Models VIII and VI: except the comparison value of Functions is 9.3024% with large difference, and the accuracy of Misc and Trees is the same, Bayes, Lazy, Meta, and Rules in Model VIII are better than Model VI.

(c) Difference of Models IX and VI: Bayes, Lazy, Meta, and Rules are better in Model IX, and the overall performance of Model IX is stable.

(d) Difference of Models X and VI: Bayes and Lazy in Model X are better than Model VI, Misc and Rules in these two models are the same, and others in Model VI are better.

(e) Difference of Models VIII and VII: Lazy, Meta, Rules, and Trees in Model VII are better than Model VIII, whereas Bayes and Functions in Model VIII are better.

(f) Difference of Models IX and VII: except Functions in Model IX is better, others in Model VII are better.

(g) Difference of Models X and VII: except Bayes and Functions in Model X are better, others in Model VII are better.

(h) Difference of Models IX and VIII: Misc in these two models is the same, only Lazy in Model IX is better, and others in Model VIII are better.

(i) Difference of Models X and VIII: Misc in these two models is the same, only Lazy in Model X is better, and others in Model VIII are better.

(j) Difference of Models X and IX: Misc in these two models is the same, only Bayes in Model X is better, and Functions, Lazy, Meta, Rules, and Trees in Model IX are better.

**Table 9.** Comparison table of highest value difference of Model VI~Model X by cross validation.

| Classify | Bayes | Functions | Lazy | Meta | Misc | Rules | Trees |
|---|---|---|---|---|---|---|---|
| Model VI | 84.3552 | 98.3087 | 77.1670 | 86.6808 | 68.4989 | 87.3150 | 89.0063 |
| Model VII | 86.8922 | 73.9958 | 99.5772 | 91.3319 | 68.4989 | 91.3319 | 89.8520 |
| Model VIII | 87.7378 | 89.0063 | 84.9894 | 88.3721 | 68.4989 | 88.5835 | 89.0063 |
| Model IX | 85.2008 | 87.5264 | 87.7378 | 87.3150 | 68.4989 | 87.7378 | 88.1607 |
| Model X | 87.3150 | 87.3150 | 87.3150 | 85.8351 | 68.4989 | 87.3150 | 87.3150 |
| Models VII and VI | 2.5370 | −24.3129 | 22.4102 | 4.6511 | 0.0000 | 4.0169 | 0.8457 |
| Models VIII and VI | 3.3826 | −9.3024 | 7.8224 | 1.6913 | 0.0000 | 1.2685 | 0.0000 |
| Models IX and VI | 0.8456 | −10.7823 | 10.5708 | 0.6342 | 0.0000 | 0.4228 | −0.8456 |
| Models X and VI | 2.9598 | −10.9937 | 10.1480 | −0.8457 | 0.0000 | 0.0000 | −1.6913 |
| Models VIII and VII | 0.8456 | 15.0105 | −14.5878 | −2.9598 | 0.0000 | −2.7484 | −0.8457 |
| Models IX and VII | −1.6914 | 13.5306 | −11.8394 | −4.0169 | 0.0000 | −3.5941 | −1.6913 |
| Models X and VII | 0.4228 | 13.3192 | −12.2622 | −5.4968 | 0.0000 | −4.0169 | −2.5370 |
| Models IX and VIII | −2.5370 | −1.4799 | 2.7484 | −1.0571 | 0.0000 | −0.8457 | −0.8456 |
| Models X and VIII | −0.4228 | −1.6913 | 2.3256 | −2.5370 | 0.0000 | −1.2685 | −1.6913 |
| Models X and IX | 2.1142 | −0.2114 | −0.4228 | −1.4799 | 0.0000 | −0.4228 | −0.8457 |

The analysis result of classifiers with higher accuracy in Models VI~X shows in Table 10. J48, LMT, and PART are used four times, respectively; IBk and Bagging are used three times, respectively; Logistic and Simple Logistic are used two times, respectively; K Star, JRip, SGD, SMO, Bayes Net, and Naive Bayes are used once, respectively. Therefore, J48, LMT, and PART are in the same times and tied for the first place, whereas IBk and Bagging tied for the second place.

In summary, the following five reports are defined from Tables 9 and 10.

(a) As the above data indicate, Model VIII with data discretization is the better model. The reason may be that the data are the information type, and the accuracy is high and good after discretization.

(b) For Model IX and Model X, the prediction accuracy of Model IX with data discretization before attribute selection is higher due to the different technology sequence selection, whereas the accuracy of Model X with attribute selection before data discretization is more stable.

(c) Compared with Model VIII with data discretization, the prediction value of Model VII with attribute selection is slightly better than that of Model VIII.

(d) The prediction values of Model VIII and Model IX with data discretization technology are not significantly different, which may be due to the fact that these attributes are all

very important, whereas Model IX with attribute selection may delete some attributes, resulting in a slight decrease in the evaluation values.

(e)   It can be seen from Table 9 that Model X is relatively stable.

(3)   Decision tree graph analysis: In this study, the percentage split (67/33) method is used to generate the decision tree, evaluate, find the optimal combination, and obtain the knowledge rules and models of decision tree, which are used as research models and provide reference for investors.

(a)   Model II: without data discretization, with attribute selection (67/33).

**Table 10.** Statistical table of classifiers with higher accuracy in Models VI~X.

| Category | Classify | Model VI | Model VII | Model VIII | Model IX | Model X | Total Times | Rank |
|---|---|---|---|---|---|---|---|---|
| Bayes | Bayes Net | | | | | V | 1 | |
| | Naïve Bayes | | | | | V | 1 | |
| Functions | Logistic | V | | | V | | 2 | |
| | SGD | | | V | | | 1 | |
| | Simple Logistic | | | V | | V | 2 | |
| | SMO | | | V | | | 1 | |
| Lazy | IBk | | V | | V | V | 3 | 2 |
| | KStar | | V | | | | 1 | |
| meta | Bagging | | V | V | V | | 3 | 2 |
| rules | JRip | | V | | | | 1 | |
| | PART | V | | V | V | V | 4 | 1 |
| trees | J48 | V | V | | V | V | 4 | 1 |
| | LMT | | V | V | V | V | 4 | 1 |

Figure 3 shows the results of decision tree of Model II in tree structure, and its interpretation of important attributes and decision tree rules of Model II without data discretization and with attribute selection (67/33) are identified.

(i)   Important attributes: E4 (marital status), E7 (total number of purchased insurance policies) and E17 (total amount of life insurance (including long-term care and disability insurance)).

(ii)   Decision tree rules: Interpreted by the first five rules.

Rule 1: IF E17 $\leq$ 30,000, E17 > 0, Then, E21 = Y.

Rule 1 explanation: If the total amount of life insurance is more than zero and less than or equal to 30,000, the policyholder would purchase the long-term care and disability insurance.

Rule 2: IF E17 $\leq$ 30,000, E17 $\leq$ 0, Then, E21 = N.

Rule 2 explanation: If the total amount of life insurance is less than zero and less than or equal to 30,000, the policyholder would not purchase the long-term care and disability insurance.

Rule 3: IF E17 > 30,000, E7 $\leq$ 1, Then, E21 = N.

Rule 3 explanation: If the total amount of life insurance is more than 30,000, and the total number of purchased insurance policies is less than or equal to 1, the policyholder would not purchase the long-term care and disability insurance.

Rule 4: IF E17 > 30,000, E7 > 1, E7 $\leq$ 2, E17 > 310,000, Then, E21 = N.

Rule 4 explanation: If the total amount of life insurance is more than 30,000, and the total number of purchased insurance policies is more than 1 and less than or equal to 2, then the total amount of life insurance is more than 310,000, the policyholder would not purchase the long-term care and disability insurance.

Rule 5: IF E17 > 30,000, E7 > 1, E7 $\leq$ 2, E17 $\leq$ 310,000, E17 $\leq$ 100,000, Then, E21 = N.

Rule 5 explanation: If the total amount of life insurance is more than 30,000, and the total number of purchased insurance policies is more than 1 and less than or equal to 2, then the total amount of life insurance is less than or equal to 310,000, or less than or equal to 100,000, the policyholder would not purchase the long-term care and disability insurance.

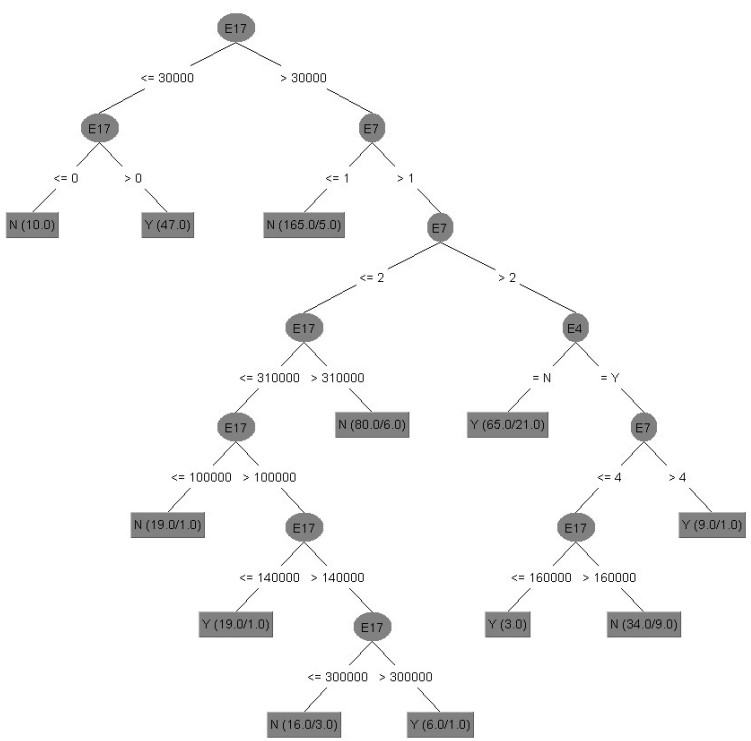

**Figure 3.** Decision Tree of Model II without data discretization and with attribute selection.

(b)  Model V: Attribute selection before data discretization (67/33).

Figure 4 shows the results of decision tree of Model V in tree structure, and its interpretation of important attributes and decision tree rules of Model V with attribute selection before data discretization (67/33) are integrated as follows:

(i)  Important attributes: E4 (marital status), E7 (total number of purchased insurance policies) and E17 (total amount of life insurance (including long-term care and disability insurance)).

(ii)  Decision tree rules (Models V and X): interpreted by all rules.

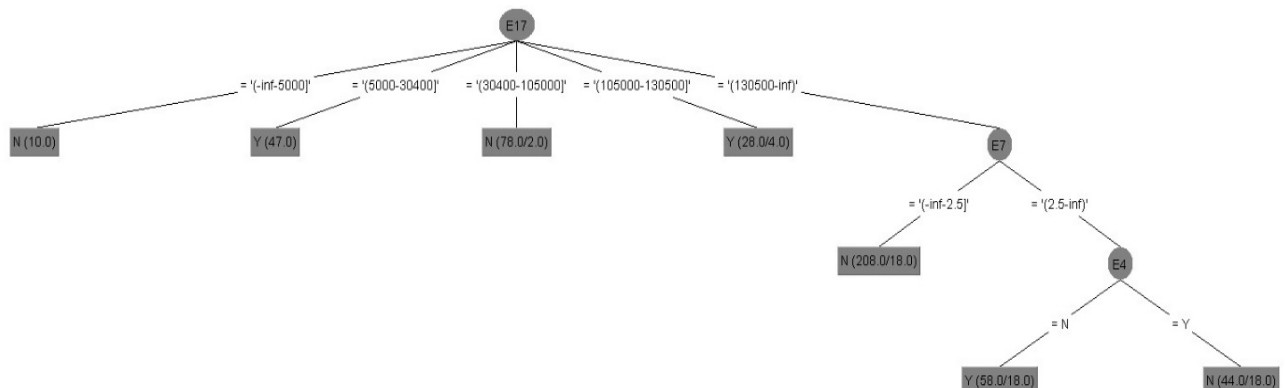

**Figure 4.** Decision tree of Model V with attribute selection before data discretization.

Rule 1: IF E17 = −inf~5000, Then, E21 = N.

Rule 1 explanation: If the total amount of life insurance is 0~5000, the policyholder would not purchase the long-term care and disability insurance.

Rule 2: IF E17 = 5000~30,400, Then, E21 = Y.

Rule 2 explanation: If the total amount of life insurance is 5000~30,400, the policyholder would purchase the long-term care and disability insurance.

Rule 3: IF E17 = 30,400~105,000, Then, E21 = N.

Rule 3 explanation: If the total amount of life insurance is 30,400~105,000, the policyholder would not purchase the long-term care and disability insurance.

Rule 4: IF E17 = 105,000~130,500, Then, E21 = Y.

Rule 4 explanation: If the total amount of life insurance is 105,000~130,500, the policyholder would purchase the long-term care and disability insurance.

Rule 5: IF E17 = 130,500~inf, E7 < −inf~2.5, Then, E21 = N.

Rule 5 explanation: If the total amount of life insurance is more than 130,500, and total number of purchased insurance policies is less than 2.5, the policyholder would not purchase the long-term care and disability insurance.

Rule 6: IF E17 = 130,500~inf, E7 = 2.5~inf, E4 = N, Then, E21 = Y.

Rule 6 explanation: If the total amount of life insurance is more than 130,500, and total number of purchased insurance policies is more than 2.5, and unmarried, the policyholder would purchase the long-term care and disability insurance.

Rule 7: IF E17 = 130,500~inf, E7 < 2.5~inf, E4 = Y, Then, E21 = N.

Rule 7 explanation: If the total amount of life insurance is more than 130,500, and total number of purchased insurance policies is more than 2.5, and married, the policyholder would not purchase the long-term care and disability insurance.

(iii) Summary: Model II and Model V are selected the same important attributes (namely E4, E7, and E17).

*4.2. Experimental Results and Findings*

Through the experiment, it finds that the evaluated accuracy with data discretization in this study is higher, most of the classifier's prediction value has been improved; the classifier prediction with both attribute selection and data discretization for processing, is relatively stable, which affirms the data pre-processing technologies, especially when two kinds of technologies have been used at the same time, making the classifier having good evaluation result.

For the decisional attribute, whether to purchase the long-term care and disability insurance, two important directions are identified and yielded, as follows:

(1) Percentage split: (a) the most stable model is Model V with attribute selection before data discretization; (b) J48 is the best classifier; (c) the accuracy of three classifiers in Trees is relatively average; (d) Misc has the worst performance.

(2) Cross validation: (a) the most stable model is Model X with attribute selection before data discretization; (b) JRip, PART, LMT, Bayes Net, Logistic, Bagging, Decision Table, and J48 are the best classifiers; (c) in Model VII, Functions decreases by 24.3129% and Lazy increases by 22.4102% at most; (d) the three classifiers in Trees are stable; (e) Misc has the worst performance.

## 5. Management Implications and Research Limitations of Empirical Results

*5.1. Management Implications*

The empirical results of this study are different from the results of previous studies in discussing the key factors of long-term care insurance, such as: the risk is an unanticipated aggregate mortality [39], a personal discretionary income is a key indicator of being insured [40], and the memory has a positive effect on the probability of owning private long-term care insurance [41]. This study adopts the method of data mining, applies the insurance data mining of existing clients of the insurance companies, constructs the prediction model to find out the little-known and hidden client groups, shorten the development time of salesmen, and achieve the accurate and time-saving business development. Issue

of purchasing the long-term care insurance policy: the problem of old age and long-term disability care is globally becoming more and more serious. In recent years, various life insurance companies have brought forth the new long-term care insurance, it is also because the market promotes the use of commercial long-term care insurance policy to solve the security problem of long-term care. Under the heavy pressure of performance and client service, the life insurance salesmen take too long to reach a deal, resulting in poor performance. In order to solve this difficulty, the big data analysis is adopted to output the important attributes after prediction, to find out the best prediction model, follow its rules to obtain the possible prior-visit list, carry out the business promotion of client demand planning, and achieve the personal performance output.

*5.2. Research Limitations*

This study finds that the pre-processing of research data is effective, and the data discretization is a very important technology in this study, which belongs to majority data and affects the evaluation performance of accuracy. The proposed method can obtain good accuracy in predicting the purchase of long-term care insurance. However, there are some limitations in the current research. (1) This study discusses the insurance industry's interest in long-term care insurance, and the research data are derived from the insurance industry. However, due to the limited number and quantity of data, most of the data are the basic information of clients, involving the personal information about salary, insurance premium, number of family members, and whether having purchased other insurance policies, so the data are also difficult to obtain, and may influence the decision factors. (2) The data used in this study is just a part of representative of the market, and it cannot completely represent all the insurance market. Therefore, it is needed to re-train and re-test the data if the proposed model is referenced and used by other time periods or other industries.

**6. Conclusions and Future Research**

This study uses the insurance industry provided data, through the data mining tools for data analysis, finds out the important condition of business work, and hidden rules. Through the designed prediction model, it finds the most helpful model and classifier, and the important research results are provided to the decision makers (insurance company and the salesman), easy to understand, and to save the salesman's time in client development. Based on the results of empirical analysis, we put forward some conclusions, discussions, and suggestions for future research.

*6.1. Conclusions*

After implementing the study framework, the empirical results are addressed for the used study data to benefit the sales confidence and corporate performance, including the following 10 key points:

(1)　Prediction models (percentage split): Models I, II, III, IV, and V. Model V with attribute selection and data discretization is better than other models and is the most stable model.

(2)　Prediction models (cross validation): Models VI, VII, VIII, IX, and X. Model X with attribute selection and data dispersion is better than other models and is the most stable model.

(3)　This study confirms that the hybrid model is better than the single model.

(4)　There is little difference in overall accuracy between percentage split and cross validation.

(5)　The classifiers with higher accuracy and lower accuracy are selected in the experiment.

(6)　Classifiers with high accuracy by percentage split are: (a) Bayes: Bayes Net, (b) Functions: Logistic, (c) Lazy: IBk, (d) Meta: Bagging, (e) Rules: PART, and (f) Trees: J48 and LMT.

(7) Classifiers with high accuracy by cross validation are: (a) Bayes: Bayes Net, (b) Functions: Logistic, (c) Lazy: IBk, (d) Meta: Bagging, (e) Rules: PART, and (f) Trees: J48.

(8) Characteristics of classifiers with poor accuracy: the overall performance of the following classifiers is exactly the same and is the worst regardless of the prediction Models I~V or VI~X. (a) Functions: SGD Text; (b) Meta: Stacking and Vote; (c) Misc: Input Mapped Classifier; (D) Rules: ZeroR.

(9) According to the decision attribute of binary classification, different prediction values are predicted in the experiment, and the same important attribute is selected. 1 decision attribute: whether to purchase the long-term care and disability insurance policy (E21). Three important conditional attributes: E4 (marital status), E7 (total number of purchased insurance policies) and E17 (total amount of insurance policies).

(10) Decision tree: it outputs the rules easy to understand, applies to insurance development in practice, and assists the salesman to obtain the innovative mode, provide better planning for clients, create a win–win–win situation of salesman, client, and insurance company. The rules from this study are summarized, and provided to decision makers as reference for judgement and important decision-making. The prediction model proposed in this study can be applied to other industries to produce different results for different practical problems.

The practical contribution of this study is illustrated in four parts: (1) the rules formed by this study results are the important and practical reference for selecting the potential clients; (2) the research is conducted on the issues concerned by the insurance industry, and the conclusions summarize the conditions for rapid selection of potential clients, so as to effectively improve the performance and achieve goals; (3) the contribution of this study is to relieve the families' financial stress and distress in taking care of the disabled; (4) the research results can be used as a reference for insurers to provide different types of commercial insurances according to individual needs for the long-term care insurance promotion, so as to make up for the missing payment items of social insurance. At the same time, it forms the five-win situation of the stakeholders, including the government, the institutions, the insurance industry, the salesman, and the public, for various purposes. In specific functions, the government can make incentives to increase interest in private insurance, and the institutions can add good supervision and regulation to the insurance sector.

*6.2. Discussion*

This study has yielded two rising directions to discuss with extended issue of long-term care insurance from perspective of hoping to pursue a good life expectancy with dignity.

First, long-term care needs are not just reserved for the elderly or sick people, and any group may suffer from accidents or illnesses. In particular, the young group encounters disability, but she/he has no good long-term care plan that is easy to fall into economic poverty. Therefore, taking out long-term care insurance can supplement the gap in nursing expenses and interrupted income in the future, and this is a really important meaning and basic context for long-term care insurance. Definitely, regarding the applicant of the long-term care insurance, it may have the upper age limit and some specific disease restrictions based on commercial and profitable consideration of insurance companies; thus, it is encouraging to give someone confidence to do this plan early. The main functions of insurance are to provide protection and certainty against future risk and accidents, offer capital, increase efficiency, and help economic progress. Thus, it is importance that minimize the economic damage caused by accidents and ensure the most basic quality of life [42]. Moreover, some limitations of this study can be minimized through a heterogeneous approach. It is suggested that subsequent researchers can analyze and integrate the databases of different life insurers to predict more diversified and effective investment portfolios. Customers can

receive more comprehensive protection through a variety of long-term care services and social policies such as National Health Insurance (NHI) in Taiwan.

Second, Taiwan is a democratization of political topic and has a broad welfare state system today [43]. Thus, discussion and analysis of the social welfare policy system or model is necessary and useful to explore and understand how social attitudes are towards the elderly or sick people and how sensitive these models are to changes under the macroeconomic and demographic situation in this country. They can be definitely addressed in the following six directions [43–45]. (1) Regarding her social welfare policy, there have six broad heading requirements for benefiting government support service, including social assistance and subsidy, social insurance, welfare service, healthcare and medicine care, employment safety, and residence justice and community building. (2) Regarding the service of healthcare and medicine care, Taiwan sets up a well-known major system of the NHI in 1996, and it can effectively take care of people's health care with low expenditure. (3) For Taiwan's elderly citizens, the National Pension Insurance (NPI) system was launched in 2008, and all non-contributory elderly have allowances under this system. (4) Moreover, the long-term care system partially offers public finance for the long-term care services began in 2015. (5) The government actively supports comprehensive family-focused policy to provide services to lessen their heavy burden of housework and workload for single parent families. (6) In recent years, due to the one of the lowest of low fertility countries in Taiwan, they have changed attitudes towards more childbearing with the incentives of maternity pension, childcare allowance, or child-raising allowance.

*6.3. Future Researches*

The financial planning market is active around the world. The mature insurance industry, financial planning, and insurance planning have become the focus of family and personal financial considerations. Insurance industry promotion focuses on the client list source. Big data analysis research is at the mature stage. The insurance company can apply the data mining technology in order to screen the client data to develop a potential client list. The application of this innovative technology is put into the scientific development technology and platform, to open up another new block for business development, and assist the salesman to save time with accurate client development. Due to the scientific and technological progress, the research tools are springing up one after another. Later researchers can use combinations of other tools for data analysis, such as: Python, R Language, SPSS, Orange, Matlab, or Artificial Neural Network (ANN), and other analytical tools. Later researchers can improve the conditional attributes and the amount of research data, and analyze the differences through the model, classifier, and accuracy evaluation of this study, to make practical application of the research results, give the client list, calculate and track the turnover rate, and then carry out the difference with the prediction accuracy. In the future, the classifier with low accuracy can be discussed to find the reasons. The management methods and decisions of insurance company managers are the biggest assistance to strengthen the support of salesmen. It is worth studying how the technological innovation can increase performance and reduce frustration.

**Author Contributions:** Conceptualization, M.-H.T.; methodology, M.-H.T. and Y.-S.C.; software, S.-F.C.; validation, Y.-S.C. and S.-F.C.; formal analysis, J.C.-L.C.; investigation, M.-H.T. and Y.-S.C.; resources, M.-H.T.; data curation, S.-F.C.; writing—original draft preparation, M.-H.T.; writing—review and editing, Y.-S.C. and C.-K.L.; visualization, Y.-S.C. and C.-K.L. All authors have read and agreed to the published version of the manuscript.

**Funding:** This research was supported by the National Science and Technology Council of Taiwan for grant numbers NSTC 110-2410-H-167-017 and 111-2221-E-167-036-MY2.

**Institutional Review Board Statement:** Not applicable.

**Informed Consent Statement:** Not applicable.

**Data Availability Statement:** Not applicable.

**Conflicts of Interest:** The authors declare no conflict of interest.

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
