# Peer review of "Application of Advanced Hybrid Models to Identify the Sustainable Financial Management Clients of Long-Term Care Insurance Policy"

_sustainability, doi:10.3390/su141912485_

Round 1

Reviewer 1 Report

This paper address an extremely relevant topic, which has become a constant concern of researchers and political decision-makers, namely, finding optimal solutions for long-term care insurance buyer by developing a prediction model. However, I also recommend to the authors to develop the concept of sustainable financial management and its benefits.

The research objectives are clearly identified. The methodology used by the authors is very complex and well demonstrated.

The authors establish three factors that are affecting the decision of choosing a long-term care insurance policy: marital status, the total number of policies purchased, and the total amount of policies.

The authors identified also some limitations of their research. Some limitations could be minimized through a heterogeneous approach to the database, in the idea of analyzing a database that would include data from several life insurers and thus a more diversified portfolio. I think this matter could be at least discussed in conclusions, along with some further recommendations for stakeholders.

Reviewer 2 Report

The reviewed article is, in my opinion, characterised by its high level of content. The article also focuses on an important and topical issue. I positively assess the reliability of the research procedure and the clearly presented limitations of the results. However, despite the overall unambiguously positive assessment, in my opinion the paper requires certain corrections and additions:

1) The aim of the study: in my opinion, it is unnecessarily repeated in the Conclusions and, what is more, it is formulated somewhat differently than in the Introduction,

2) The extent to which the reviewed paper and the conclusions formulated by its authors fill a research gap needs to be clearly and convincingly justified. In this context, it would be useful to refer to at least a few articles or monographs in which issues related to those described in the article are addressed. Referring to what has already been demonstrated and the defined research gap, it would be worthwhile to formulate research hypotheses,

3) The article lacks, in my opinion, a fair discussion of the results by contrasting them with the research results of other authors (to what extent the conclusions overlap, to what extent they do not overlap and why this is the case),

4) In the title of the article and then only perhaps 3 times in the text the term "Sustainable Financial Management Clients of Long-Term Care Insurance Policy" appears. The term is not explained (not defined), especially the context of the use of the term 'Sustainable',

5) It would have been worthwhile to clarify the description of the source data: how many clients the sample consisted of, why such and not a different period was chosen, whether the sample is representative of the market, whether the insurance company from which the data is taken is representative, etc...,

6) It would be useful to add a commentary on how much the models constructed are dependent on the welfare and social policy model in the country, social attitudes towards the elderly or sick, and how sensitive these models are to changes in the macroeconomic and demographic situation,

7) In my opinion, among the stakeholders for whom the conclusions of this study would be important should be the institutions that supervise and regulate the insurance sector and the government (with a view to the possible introduction of incentives to increase interest in private insurance),

8) I wonder whether the implementation of the study's practical conclusions for insurance companies and their representatives will increase the exclusion of elderly and sick people from the pool of potential customers of insurance companies. Please give arguments I am wrong. 

The above comments are polemical in nature and do not affect the overall high assessment of the article. 
